# Optimization, Not Architecture, Governs Vision Transformer Generalization in Small-Data Regimes

**Divyanshu Gupta**
Independent Researcher
`divyanshugupta1331@gmail.com`

## Abstract

Vision Transformers (ViTs) perform competitively on large-scale vision benchmarks but consistently underperform convolutional models when trained from scratch on small datasets. We present a controlled empirical study of ViTs trained from scratch on CIFAR-10, systematically isolating the effects of data diversity, model capacity, regularization, and optimization. Across four progressively refined ViT variants, we find that architectural scaling and data augmentation yield limited gains, whereas optimization strategies—specifically learning rate warmup and cosine decay combined with stronger regularization—produce substantial improvements in generalization. Our results indicate that ViT failure in small-data regimes is governed primarily by optimization dynamics rather than architectural limitations.

## 1 Introduction

Vision Transformers (ViTs) replace convolutional inductive biases with global self-attention and achieve strong performance when trained on large-scale datasets (Dosovitskiy et al., 2021). However, when trained from scratch on small or low-resolution datasets, ViTs consistently underperform convolutional neural networks and exhibit unstable generalization behavior (Touvron et al., 2021). This limitation is often attributed to data inefficiency or the absence of built-in spatial priors, yet these explanations conflate multiple interacting factors (Zhang et al., 2017).

In practice, ViT performance in small-data regimes depends on several tightly coupled elements, including data diversity, architectural capacity, regularization, and optimization dynamics. Isolating the contribution of each factor is necessary to understand whether ViT failures arise from fundamental architectural limitations or from correctable training deficiencies.

In this work, we adopt a controlled, scientific-method-driven approach to study Vision Transformer generalization on CIFAR-10. We train ViTs from scratch while modifying a single factor at a time, examining the effects of data augmentation, architectural scaling, regularization, and learning rate scheduling on training dynamics and validation performance. This design enables causal interpretation of observed performance changes.

Our findings align with prior large-scale studies on Vision Transformer training dynamics, but we isolate these effects in controlled small-data regimes to better understand their causal role.

Our results reveal a consistent pattern: data augmentation and architectural scaling provide only incremental improvements, whereas optimization strategies—particularly learning rate warmup followed by cosine decay—produce a qualitative shift in learning behavior, enabling sustained late-stage generalization. We further compare ViTs against a convolutional baseline trained under identical conditions, highlighting the role of inductive bias while demonstrating that optimization substantially narrows the performance gap.

**Contributions.** We (1) provide a controlled empirical characterization of ViT failure modes in small-data regimes; (2) show that architectural scaling and data augmentation alone yield limited

generalization gains; and (3) identify optimization dynamics as the primary driver of ViT performance when training from scratch on limited data.

## 2 EXPERIMENTAL SETUP AND METHODOLOGY

### 2.1 DATASET AND TRAINING REGIME

We conduct all experiments on CIFAR-10, a widely used small-scale image classification benchmark containing 50,000 images across 10 object categories at a spatial resolution of $32 \times 32$ pixels. This dataset provides a representative small-data setting in which Vision Transformers (ViTs) are known to underperform convolutional architectures when trained from scratch.

To study generalization behavior under limited data, we employ two controlled split configurations. For the convolutional baseline (VGG16) and the first ViT variant, we use a 40k/10k train–test split. For subsequent ViT variants, we increase the effective training set to 49k images and reserve 1k images as a held-out evaluation set. All training is performed exclusively on the training subset, and performance is measured on the corresponding held-out split, which serves as both validation and testing data.

All models are trained from scratch without pretraining or transfer learning. Training protocols are kept consistent across experiments to ensure fair comparison. In particular, all ViT models use a batch size of 64, the Adam optimizer, cross-entropy loss, and early stopping based on validation accuracy. The VGG16 baseline uses sparse categorical cross-entropy loss under the same evaluation criteria.

### 2.2 BASELINE VISION TRANSFORMER ARCHITECTURE

Our baseline model (Model-1) follows a compact Vision Transformer architecture adapted to CIFAR-10 resolution. Input images are partitioned into non-overlapping patches using a Conv2D-based patch embedding layer, where the kernel size and stride equal the patch size, enabling efficient patch extraction directly on the GPU. The resulting patch embeddings are combined with learned positional embeddings, and a learnable classification token is prepended to the sequence.

The sequence is processed by a stack of transformer encoder layers consisting of multi-head self-attention, feed-forward networks, layer normalization, and residual connections. The output corresponding to the classification token is passed to a linear classifier to produce class logits.

To reflect realistic small-data constraints, we intentionally adopt a compact configuration rather than scaling to large models that would trivially overfit. Importantly, the baseline ViT is trained without any data augmentation, regularization, or learning rate scheduling, serving as a minimal reference for subsequent controlled improvements.

### 2.3 SCIENTIFIC METHOD AND CONTROLLED DESIGN

The central goal of this study is to identify which factors most strongly influence Vision Transformer generalization when training data is limited. Rather than jointly tuning many components, we adopt a controlled, progressive experimental design in which modifications are introduced incrementally across successive model variants.

Starting from the baseline ViT, we progressively incorporate: (i) increased data diversity through augmentation, (ii) greater model capacity together with dropout-based regularization, and (iii) improved optimization dynamics through learning rate warmup and cosine decay scheduling (Goyal et al., 2017; Loshchilov & Hutter, 2017). This staged design enables clearer attribution of performance changes to specific training factors.

Concretely, the four variants correspond to: a baseline ViT without augmentation, a version with data augmentation only, a higher-capacity model with added regularization, and a final variant that combines stronger augmentation with an improved learning rate schedule. Rather than optimizing for state-of-the-art accuracy, our objective is to analyze training behavior, generalization trends, and failure modes under progressively refined conditions.

## 3 CONTROLLED EXPERIMENTS AND OBSERVATIONS

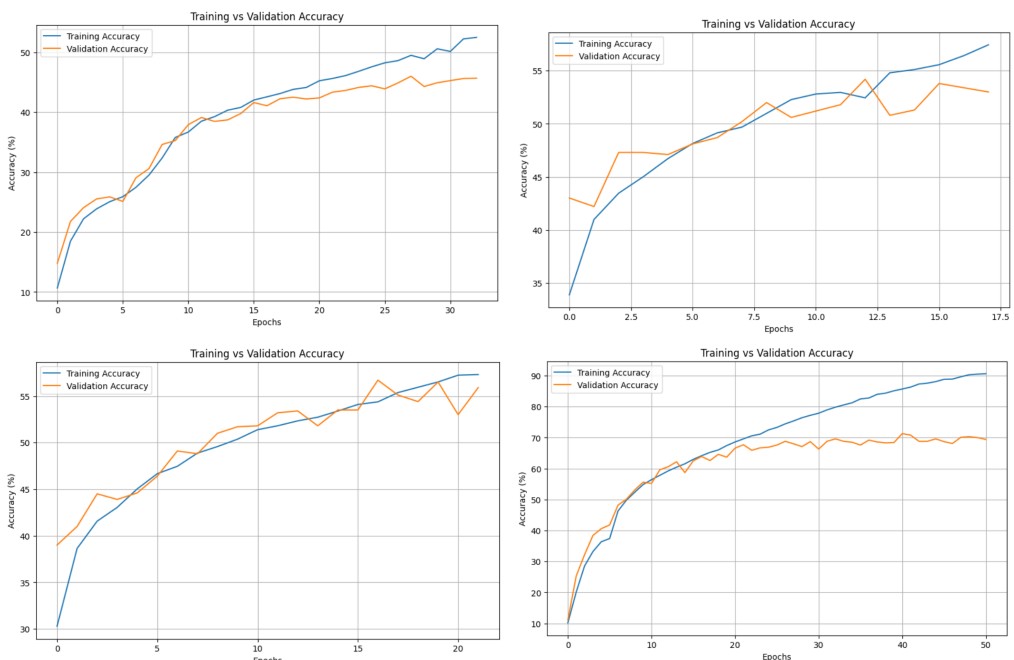

Figure 1: Validation accuracy trajectories for four Vision Transformer variants trained on CIFAR-10. Models progress from (top-left) baseline ViT, to (top-right) data augmentation, (bottom-left) increased capacity with regularization, and (bottom-right) optimized training with learning rate warmup and cosine decay. The first three variants exhibit early validation plateaus with only incremental improvements, whereas the optimized training regime shows sustained late-stage gains and substantially higher generalization performance, highlighting the dominant role of optimization dynamics in small-data settings.

### 3.1 ARCHITECTURAL AND DATA-CENTRIC MODIFICATIONS

We begin with a baseline Vision Transformer trained from scratch on CIFAR-10 without data augmentation, explicit regularization, or learning rate scheduling. Although training accuracy improves steadily, validation performance plateaus early and exhibits a widening generalization gap, indicating weakly discriminative representations and overfitting.

Introducing standard data augmentation increases training diversity and improves early validation accuracy, raising peak performance from approximately 45% in the baseline to about 54%. Increasing model capacity through deeper transformer stacks and additional attention heads, combined with dropout-based regularization, provides only modest additional gains, improving validation accuracy further to roughly 56–57%. Despite these architectural and data-centric modifications, validation accuracy continues to plateau well before training accuracy converges, indicating diminishing returns from scaling capacity alone in small-data regimes.

Overall, these results show that augmentation and architectural scaling provide incremental improvements, but do not fundamentally alter the learning dynamics of attention-based models trained from scratch on limited data.

### 3.2 OPTIMIZATION DYNAMICS

In contrast, introducing learning rate warmup followed by cosine decay produces a qualitatively different learning regime (Goyal et al., 2017; Loshchilov & Hutter, 2017; Steiner et al., 2022). Rather than plateauing early, validation accuracy improves steadily over extended training and continues to rise well into later epochs. This optimized training strategy increases peak validation performance

substantially to approximately 71%, representing a gain of more than 14 percentage points over the previous variant.

This shift demonstrates that optimization dynamics exert a substantially larger influence on Vision Transformer generalization than architectural or data-centric modifications alone. Stable early optimization and gradual learning rate decay enable sustained representation refinement, allowing the model to break through the performance ceiling observed in earlier variants.

## 4 INDUCTIVE BIAS COMPARISON: VIT VS CNN

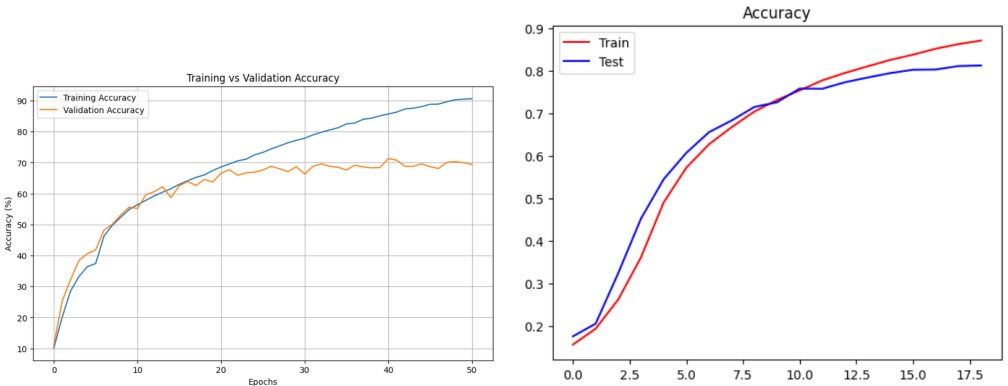

Figure 2: Validation accuracy comparison between the optimized Vision Transformer (left) and a VGG16 convolutional baseline (right), trained from scratch on CIFAR-10.

To contextualize ViT behavior, we compare the optimized Vision Transformer against a VGG16-style convolutional network trained from scratch under comparable data splits, epochs, and evaluation metrics. The convolutional model converges faster and achieves higher validation accuracy (approximately 85–90%), whereas the ViT attains lower peak performance (approximately 70–71%) and remains more sensitive to optimization choices.

This gap reflects architectural inductive bias: convolutions encode locality and translation invariance directly, enabling efficient learning from limited data, while Vision Transformers must learn such structure implicitly through data and optimization (LeCun et al., 1998). Although improved training substantially narrows the gap, CNNs retain a clear advantage in small, low- resolution regimes.

Taken together, these findings suggest that the primary limitation of Vision Transformers in small-data settings is not insufficient model capacity but weaker inductive bias. When training data is scarce and resolution is low, convolutional priors provide strong structural guidance that enables efficient generalization, whereas transformers rely more heavily on optimization and data diversity to learn comparable structure. Consequently, careful optimization can mitigate—but not fully eliminate—the inherent data efficiency advantage of convolutional architectures.

## 5 CONCLUSION

We presented a controlled empirical study of Vision Transformers trained from scratch on CIFAR-10. Across progressively refined variants, architectural scaling and augmentation yielded incremental gains, whereas optimization strategies—particularly warmup and cosine decay—produced the largest improvements in generalization. These results indicate that optimization, more than architecture, governs ViT performance in small-data regimes. These observations contribute toward a clearer scientific understanding of optimization-driven generalization behavior in attention-based models under data-constrained settings.

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

## A    EXPERIMENTAL DETAILS AND HYPERPARAMETERS

Table 1 summarizes the architectural and optimization hyperparameters used across Vision Transformer experiments. Parameters are kept constant unless explicitly modified for a given variant.

Table 1: Training and architectural hyperparameters for ViT experiments.

| Parameter | Value |
|---|---|
| Dataset | CIFAR-10 |
| Image resolution | $32 \times 32$ |
| Train/test split | 40k/10k (Model-1), 49k/1k (Models 2–4) |
| Patch size | $8 \times 8$ |
| Number of patches | 16 per image |
| Embedding dimension | 192 |
| Transformer encoder blocks | 2 (Models 1–2), 8 (Models 3–4) |
| Attention heads | 6 (Models 1–2), 12 (Models 3–4) |
| Optimizer | Adam |
| Loss function | Cross-entropy |
| Batch size | 64 |
| Learning rate | 0.001 (Models 1–3) |
| Learning rate schedule | Warmup + cosine decay (Model-4 only) |
| Warmup LR range (Model-4) | $1 \times 10^{-5} \to 3 \times 10^{-4}$ |
| Warmup epochs | 5 (Model-4 only) |
| Dropout | 0.1 (Models 3–4) |
| Weight decay | None |
| Max epochs | 200 (Models 1–3), 100 (Model-4) |
| Early stopping patience | 5 epochs (Models 1–3), 10 epochs (Model-4) |

