# OpenReview forum: "Optimization, Not Architecture, Governs Vision Transformer Generalization in Small-Data Regimes"
_ICLR.cc/2026/Workshop/Sci4DL — Sci4DL 2026_

### Official Review · Reviewer_AxiY · 2026-02-10

**Fit:** 3
**Significance:** 2
**Confidence:** 2

**Summary:**

This paper attempts to tackle the question: why do vision transformers (ViTs) perform relatively poorly when the dataset size is small? To evaluate this, they use a basic training configuration for a ViT with absolutely no regularization, augmentation, constant learning rate with Adam, etc., and train on CIFAR10 classification as a baseline. They then add augmentations, dropout, and linear warmup-cosine decay learning rate scheduling (separately at first, then combining them together) and observe that improving the optimization dynamics is the most critical for obtaining strong validation performance (test accuracy). Finally they show empirical evidence that a CNN trained on the merged-improvement setup performs better than the equivalently sized transformer under fair comparison, highlighting the role of inductive bias at small scales.

**Strengths:**

- The question of "what causes transformers to perform badly at small scale" is interesting; it seems to be common intuition, as remarked in the paper, that the cause is lack of inductive bias, but it is good to rule out any competing hypotheses such as "bad optimization" or "bad augmentation pipeline".

**Suggestions:**

- The paper may want to tackle the issue of compositionality more carefully. As I understand, the four settings tackled (baseline, adding augmentation, better regularization with dropout, better optimization with cosine decay LR schedule) build off each other: each stage contains all the improvements of the last stages. It may prove the point better to study each effect independently as well as compositionally, instead of attributing all the success to the LR schedule (when in reality it may be the composition of LR schedule and dropout, for example).

- The paper may want to tackle more completely the question of how (mechanistically) the inductive bias helps the most. E.g., if convolutions help, how much do they help and how well can they compose with other standard networks? Maybe a hybrid CNN-transformer may perform the best, with the best architecture varying the (# conv layers) / (# attention layers) ratio  at different scales?

---

### Official Review · Reviewer_oyJf · 2026-02-27

**Fit:** 2
**Significance:** 1
**Confidence:** 2

**Summary:**

The paper conducts an empirical study on training Vision Transformers (ViTs) from scratch on the CIFAR-10 dataset. By progressively adding data augmentation, model capacity/dropout, and finally a learning rate warmup with cosine decay, the authors conclude that optimization dynamics play a larger role in ViT generalization on small datasets than architectural scaling. They also compare the final model to a VGG16 baseline to illustrate the ongoing performance gap caused by the ViT's lack of convolutional inductive biases.

**Strengths:**

Here are the primary strengths of the work:

- Methodology is clear and has a controlled, step-by-step design to isolate potential confounding aspects.
- Narrative is easy-to-follow and the language is clear.

**Suggestions:**

- The paper in its current form frames the impact of learning rate warmup and cosine decay as a novel contribution/insight for training ViTs on small data. However, these optimization strategies, along with heavy augmentation, are widely documented as standard prerequisites for ViT training in foundational literature ([A, B]). The authors should more explicitly discuss these prior works and clarify what new scientific understanding this study provides beyond confirming established practices.
- The progressive design introduces a confounding factor in the final stage. Model-4 introduces the improved learning rate schedule, but it applies this to a model that already includes increased capacity, dropout, and augmentation. To cleanly support the claim that "optimization, more than architecture, governs ViT performance", the authors should include an ablation study that applies the optimized learning rate schedule directly to the baseline architecture
- Given the established nature of the recipes utilized by the work, the work might be better positioned as an empirical reproduction of well-known ViT training dynamics, rather than having a novel/interesting contribution on ViT training dynamics.


---
[A] Touvron, Hugo, et al. "Training data-efficient image transformers & distillation through attention." International conference on machine learning. PMLR, 2021.

[B] Steiner, A., Kolesnikov, A., Zhai, X., Wightman, R., Uszkoreit, J., & Beyer, L. (2021). How to train your vit? data, augmentation, and regularization in vision transformers. arXiv preprint arXiv:2106.10270.

---

### Meta-Review · Area_Chair_35no · 2026-02-28

**Recommendation:** Accept

**Metareview:**

This paper shows that even at small scale, optimization hyperparameters play a dominant role in the performance of ViTs and enables narrowing (but not closing) the gap with CNNs. I encourage the authors to follow the reviewers' suggestions to discuss the relation with prior work (done at larger scales) in greater depth and include additional experiments to strengthen the main claim. I recommend acceptance.

---

### Decision · Program_Chairs · 2026-03-02

Accept